# Cognitive Training Therapy Improves the Effect of Hypocaloric Treatment on Subjects with Overweight/Obesity: A Randomised Clinical Trial

**DOI:** 10.3390/nu11040925

**Published:** 2019-04-24

**Authors:** Joaquín S. Galindo Muñoz, Juana María Morillas-Ruiz, María Gómez Gallego, Inmaculada Díaz Soler, María del Carmen Barberá Ortega, Carlos M. Martínez, Juan José Hernández Morante

**Affiliations:** 1Endocrinology and Nutrition Service, Miguel Servet Hospital, 50009 Zaragoza, Spain; joaquinsantiago.gm@gmail.com; 2Food Technology & Nutrition Dept., Catholic University of Murcia, 30107 Murcia, Spain; jmmorillas@ucam.edu; 3Faculty of Medicine, Catholic University of Murcia, 30107 Murcia, Spain; mggallego@ucam.edu; 4Faculty of Nursing, Catholic University of Murcia, 30107 Murcia, Spain; inmadiaz93@hotmail.es (I.D.S.); mcbarbera@ucam.edu (M.d.C.B.O.); 5Biomedical Research Institute (IMIB), Arrixaca University Hospital, 30120 Murcia, Spain; cmmarti@um.es

**Keywords:** cognitive training, executive function, weight loss, clinical trial, decision-making

## Abstract

Obesity has been associated with impaired cognitive performance. This study aimed to determine whether improvements in cognitive function may contribute to higher weight loss in patients with obesity. In this randomised, 12-week trial, participants with overweight/obesity were randomised into a cognitive training intervention (Cognitive) group or a cognitive-behavioural (Control) group. In addition, both groups followed a hypocaloric dietary treatment. Cognitive functioning measurements and anthropometrical parameters were evaluated. All cognitive measures improved in the intervention group (*p* < 0.005 in all contrasts). In controls, significant improvements in attention, flexibility and task planning were also observed. Regarding anthropometrical parameters, the effect of the intervention in the cognitive group was higher for the total percentage of weight loss, body mass index (BMI), body fat and waist circumference. Biochemical parameters improved in both groups. Attending to our data, cognitive training was more effective that the hypocaloric intervention alone, partly related to an improvement in the working memory. Despite the shortage of training interventions for executive functions in the context of weight control, this type of combined intervention could establish the first steps towards a more appropriate intervention for patients with obesity.

## 1. Introduction

The interaction between obesity and cardiovascular diseases is well established, and recent observations seem to indicate that obesity is also associated with poorer cognitive performance, cognitive decline and dementia [1]. Executive functions, the higher-level cognitive processes, are especially affected in patients with obesity [2,3]. Previous studies have found poorer performance in several executive functions, like inhibition, flexibility and decision making, whereas the interaction between obesity and other functions, such as working memory, planning and reasoning, is not evident [4,5,6].

The effect of certain appetite-regulating hormones on the function of the prefrontal cortex may explain the relationship between obesity and cognitive decline. Ghrelin, an orexinergic hormone that exists in increased levels in subjects with elevated body weight, signals on prefrontal cortex receptors [7]. Moreover, animals lacking glucagon-like peptide-1 (*glp1*^−\−^), an anorexigenic hormone, show memory deficits [8]. Other satiety hormones, like leptin, modulates hippocampal synaptic plasticity [9]. The leptin and/or glp1-resistance present in subjects with obesity may, therefore, explain certain deficits in hippocampal-dependent learning and memory [10,11]. Other processes like a chronic low-grade proinflammatory mediators (TNF-α and several interleukins) have been known to produce cognitive deficits in humans [12]. 

However, it is unclear whether weight loss is able to improve cognitive performance. A previous trial performed by Espeland et al. showed that intensive lifestyle intervention did not alter cognitive function in adults with obesity and type 2 diabetes [13]. In contrast, the study of Horie et al. showed an association between body mass index decrease and improvements in verbal memory, executive function, and global cognition [14].

Cognitive deficits in those patients with elevated body fat may be the result of the metabolic impairment related to obesity, and, cyclically, may also be themselves a risk factor for developing obesity. Deficits in impulse control are reported to be a key mechanism for obesity development [15] and poorer executive functions are related to less physical activity [16]. In this regard, it is tempting to speculate that an executive functioning deficit may induce worse eating and less healthy habits, which in the long term may favour obesity development. Therefore, improving executive functions may increase the ability to regulate impulsive behaviours leading to obesity. In fact, recent reports have showed that, in children, computer training of executive functions may improve the effectiveness of weight loss programs [17].

Consequently, this study aimed to determine whether improvements in cognitive function may contribute to higher weight loss in patients presenting overweight/obesity. To that end, we assessed the effect of adding cognitive training therapy to the usual weight loss hypocaloric dietary treatment.

## 2. Materials and Methods

### 2.1. Design 

This randomised clinical trial was conducted from January–December 2017 at Catholic University of Murcia facilities. Written informed consent was required from each patient for participation in the study. The protocol of this randomised trial adheres to the CONSORT guidelines [18]. The CONSORT checklist is available in Appendix A. To evaluate the impact of cognitive training therapy on weight loss, a double-blind (de facto masking) trial was designed. Patients were informed that the objective of the study was to evaluate the relationship between obesity and cognition, but did not know the different intervention possibilities. 

One researcher (J.J.H.M.) carried out the randomisation with the assistance of a macro designed in Visual Basic for Microsoft Excel. Randomisation divided the participants into two groups, depending on whether they were treated with a hypocaloric diet (CONTROL group), or the group that followed a hypocaloric diet plus 12 cognitive training sessions (COGNITIVE group) of intervention. In order to obtain a similar size in both groups, a randomisation in blocks with a 1:1 allocation ratio was performed.

One week before the beginning of the cognitive and dietary intervention, the participants performed a series of cognitive tests, as described below. After 12 weeks of intervention, all participants were re-examined to measure their performance on the neurocognitive tests and to evaluate possible changes in anthropometrical parameters.

### 2.2. Participants

The sample size required for the study was determined with the help of the GPower 3.0 program [19]. The sample size was estimated using a two-sided F-test with a significance level of 95% and considered a statistical power (β) of 80%, as well as a between-group treatment effect difference (d) of 10 kg, which may represent an average 10% weight loss, which has been stated as enough to promote a significant health improvement in these patients [20]. A standard deviation (σ) of 10 kg was assumed, following our previous work conducted in the same setting [21]. This procedure designated a total of 30 subjects per group. Taking into account a withdrawal ratio of 25% (also based on the previous study), the final minimum sample for the present study was 40 subjects per group. Figure 1 shows the flow diagram for the selection of the subjects of this study. 

Selection criteria included having a body mass index (BMI) ≥ 27 kg/m^2^ and volunteering to be a part of the study. Those subjects with type 2 diabetes, cardiovascular event (stroke, acute myocardial infarction, intermittent claudication, etc.) antecedents, kidney or liver failure and/or any other significant or chronic condition known to affect cognitive status (depression, cancer, etc.) were excluded from the study. Those subjects using some sort of pharmacological treatment that could affect body weight (corticoids, thyroxine, etc.) were also excluded. In addition, to be on a dietary treatment or diet at least six months before participation in the study was also considered an exclusion criterion. The Mini-Mental Status Examination test was conducted to exclude subjects with a possible cognitive decline [22]; however, no participant showed a score indicating cognitive impairment. Finally, 120 participants (74% women) composed the initial study sample. 

The present work was carried out with previous written authorisation by the Catholic University of Murcia’s Ethics Committee. Patients were informed about the design of the study orally and in written form. Consent to be part of the study was also requested in both ways. An explanation of the research project in the ethical sense was also given, informing them about the aim of the results obtained, warranting confidentiality and anonymity of the data and respecting the Helsinki Declaration Agreement and local legislation on biomedical research (Clinical Trial Information: Name: Cognitive Therapy Plus Dietary Intervention for Obesity Treatment (COGNI-OB). Registration Number: NCT03749772. Available at: http://clinicaltrials.gov).

### 2.3. Outcome Measures

As recommended by the International Federation for the Surgery of Obesity and Metabolic Disorders (IFSO) and SEEDO [20,23], the primary efficacy end-point was change in the percentage of total weight loss (%TWL) from baseline to week 12, rather than changes in body weight only. Secondary efficacy end-points included changes in anthropometrical (BMI, body fat and waist circumference) and biochemical parameters (fasting glucose, cHDL and triglycerides) from baseline to week 12. The change in executive function performance was also considered a priori as a secondary end-point measure of treatment effectiveness.

### 2.4. Intervention

The participants of the cognitive group performed cognitive training therapy during 12 weekly sessions, carried out during the three months of hypocaloric treatment. Cognitive training was conducted via the video game Brain Exercise^TM^ (Bandai Namco Games Ltd., Tokyo, Japan). The participants completed 12 different practice exercises in every session. The exercises included in these sessions were based on sorting (e.g., classifying subjects attending to a rule), memory (e.g., remembering the precise location of an object), calculation (e.g., mathematical problems), problem-solving, logical and similar games or exercises. The time employed to perform these exercises in every session was 30 min. Brain Exercise^TM^ was selected because it is one of the most popular memory training games in the world (it is the PC version of the Brain Training^TM^ video game for the Nintendo DS console, Nintendo Co. Ltd., Kyoto, Japan). More importantly, this video game was selected for its efficacy, as it has been developed with a high level of neuropsychological evidence [24,25], and previous studies have shown its effectiveness in increasing certain executive functions [26,27] (detailed information is available in Appendix A). The control group followed 30 min sessions of cognitive-behavioural therapies and nutritional education, performed during 12 sessions. These sessions were conducted to ensure a similar duration of time and number of sessions than cognitive group. 

### 2.5. Hypocaloric Dietary Treatment

All patients with overweight/obesity, from both the Intervention and Control groups, followed a dietary treatment for three months (12 weeks). Hypocaloric diets were designed following the SEEDO-FESNAD guidelines [20]. Patients were instructed to substitute their usual diet for a balanced diet, following a nutrient distribution based on the Mediterranean Diet. The patients were monitored weekly to record their changes in weight and body composition. From these data, resting metabolic rate and daily energy requirements (using an ‘average’ physical activity factor of 1.3) were estimated. In addition, participants were recommended to get 30 min of moderate aerobic exercise for at least five days per week (150 min/week) [20].

The dietary treatment was performed through diets structured in five meals with the following caloric distribution (±1%): breakfast 20%, mid-morning snack 10%, lunch 35%, mid-afternoon snack 10% and dinner 25%. The diets were designed according to the patients’ preferences, and unwanted meals were excluded from the menus. Daily energy expenditure was recalculated weekly, and the diets were reformulated on this same basis to try to more precisely adjust to the patient’s energy requirements. Dietitian provided new menus every week based on the actual patients’ energy expenditure. All diets were designed with the assistance of the Dietowin 7.0 software (Bl-Biologica, Barcelona, Spain).

### 2.6. Measures 

#### 2.6.1. Anthropometric and Clinical Data

Anthropometric variables were evaluated according to the criteria proposed by the SEEDO in 2007 [28]. The parameters of weight and percentage of fat mass were measured by bioelectrical impedance analysis, with a TANITA MC-780^®^ (TANITA Corporation of America, Inc., Arlington Heights, IL, USA). The height was measured with a TANITA rod (model Harpender). The patient’s BMI was determined from these data. Body fat distribution was analysed with the measurement of waist circumference. Each measurement was performed three times, in a non-consecutive way, by the same investigator. The biochemical parameters of cHDL, triacylglycerides (TG) and fasting basal glucose were analysed with the Reflotron^®^Plus (Roche Diagnostics, Basel, Switzerland) after an intravenous blood sample extraction at the beginning and at the end of the study.

#### 2.6.2. Executive Function Assessment

The evaluation of the patients’ executive function performances was carried out through the following exercises or cognitive tests:

Working memory–letter–number sequencing [29]: This test is a component of the Wechsler Adult Intelligence Scale (3rd Edition), designed to measure working memory, attention and sequential abilities. Patients are read a random sequence of combined letters and numbers, and, subsequently, they are asked to repeat the sequence, first placing the numbers in ascending order and then the letters in alphabetical order. The interest variable is the number of correct answers, with a score ranging from 0–21 points.

d2 attention test [30]: This test measures cognitive processes, such as attention, mental concentration or attentional control. The test consists of 14 lines with 47 characters each, with the letters ‘d’ and ‘p’ having one or two dashes on the top or bottom of the letter. The subject may carefully check each line, from left to right, and mark any ‘d’ character with two dashes (either the two dashes above, below or one above and one below). The main variables are the total test effectiveness, measured as the total number of responses minus the sum of omissions and commissions.

5-digit test [31]: This test is a simple alternative to the Stroop test, with the advantage that it is independent of the language and cultural level of the participant. It is based on reading digits, from 1–5, in a series of 50 images, combined in a different way. Two complementary variables are then obtained: inhibition, the time of choice minus the time reading, and mental flexibility, the time of alternation minus the time of reading. 

Rey–Osterrieth complex figure test [32]: The Rey–Osterrieth figure is reproduced in order to evaluate perceptual organisation and task planning. The procedure consists of making a copy of the figure, which is presented to the subjects, with the model in view (copy phase). The copy is made by hand and has no time limit. Three minutes later, the subject must reproduce the figure without the help of the model, then again after thirty minutes (also by memory). In order to obtain an assessment of the test, the figure is divided into 18 parts, each of which is given a score. The maximum score is 36 points. 

Iowa Gambling Task (IGT) [33]: This task evaluates decision-making function. It has been developed into a kind of video game, where four decks of cards, labelled with the letters A, B, C and D, are presented to the subject, who must freely choose one of the four decks. Of the four options, decks A and B provide higher economic rewards but also higher penalties, whereas decks C and D provide lower rewards but also lower penalties; however, the subject does not know all this. The objective of this test is to win as much money as possible. The test has a duration of 100 card choices, after which the test ends. 

The Gambling Index (GI) is obtained by subtracting the total of choices from the disadvantageous decks to the number of choices of the advantageous decks: (C + D) − (A + B). 

### 2.7. Statistical Analysis

First, a basic descriptive statistical analysis was performed to evaluate the general characteristics of the study population. In order to compare the different executive functions with each other, attending to the reference norms of cognitive change in Spanish old adults, the data were typed in z-scores following the formula: z-score = (x − µ)/σ, where x denotes the patient’s score in each test, µ is the normalised mean value of a similar population and σ represents the standard deviation of the normalisation population. The µ and σ values were obtained from the manufacturer’s cognitive test instructions [32]. 

The efficacy analyses were performed on the data from the full analysis set, which included all randomised participants that completed baseline and final evaluations and did lost less than two intervention sessions. As recommended by ICH guidelines, the presented results are based on the last observation carried forward imputation unless otherwise noted. The robustness of the primary analysis was investigated by multiple sensitivity analyses using other methods for handling missing data [34], confirming the reliability of the data obtained with the primary analysis. 

Baseline characteristic differences between both groups were analysed by means of a Student’s t-test. To evaluate the change of anthropometrical parameters at the end of the treatment compared to baseline values (Δ parameter = final value − baseline value), an ANCOVA analysis was performed to assess possible estimated treatment effect differences between groups. In the same way, the same a priori ANCOVA analysis was carried out in order to exclude possible bias due to age, sex and education level. Baseline anthropometric characteristics were also considered as a covariate.

All statistical tests were performed considering a significance level of *p* < 0.050. The analysis was carried out with the help of the software SPSS for statistical analysis (22.0.7 release, SPSS Inc., Chicago, IL, USA). 

## 3. Results

### 3.1. Sample Characteristics

The final sample consisted of 120 subjects, whose demographic and clinical characteristics are shown in Table 1. Subjects were randomly allocated into trial groups (cognitive and control). The allocation rate was 1:1. Of the 120 randomised participants, 96 (cognitive: 48 [80%], control: 48 [80%]) completed the trial (Figure 1). There was no significant difference between groups in withdrawal rate (χ^2^ = 0.677, *p* = 0.537). A higher proportion of subjects withdrew by the lack of adherence to the intervention, while the drop-out by the loss of baseline or final evaluations was small.

The groups were comparable regarding demographic and basal cognitive scores (Table 1). The detailed information regarding Iowa gambling task-blocks performance is available in Appendix A. Both groups were similar in terms of age, sex, study level and other clinical characteristics; although, plasma cHDLs were slightly higher in the cognitive group. This similarity was expected, considering the random selection of the subjects in each group.

### 3.2. Efficacy of Cognitive Training on Executive Functions

Results from ANCOVA analyses showed that all cognitive measures improved in the cognitive group (*p* < 0.005). In the control group, significant improvements in selective attention, flexibility and task planning were observed (Table 2). Between-group differences in improvement of cognitive functions only reached statistical significance for working memory and task planning (Table 2). This significance was still observed after controlling for age, baseline cognitive characteristics and study level. 

### 3.3. Efficacy of Cognitive Training on Anthropometrical and Biochemical Parameters

In both groups, all anthropometrical parameters improved after three months (Figure 2), but the estimated treatment effect in the cognitive group was statistically and significantly higher for the total %TWL, as well as for BMI, body fat and waist circumference (Figure 2). These differences remain statistically significant after adjusting for baseline anthropometrical values and sex, as defined in the pre-specified ANCOVA analysis (Figure 2). To reinforce this observation, the different statistical procedures conducted in the sensitivity analysis confirmed this slight but significant effect of the cognitive treatment. 

Concerning biochemical parameters, at three months, both TG and cHDL significantly improved in both the cognitive and control groups (Figure 3), which indicates a similar treatment effect in both groups. After three months, clinical characteristics were mainly similar in both groups, probably as a consequence of the baseline values; although, cHDL were still higher in the cognitive group (Figure 3).

## 4. Discussion

Nowadays, it is well established that there is a relation between cognitive deficit and obesity development [35]; however, it has not been possible to establish a causal relationship between these variables [36]. In this regard, the aim of this study was to verify whether the improvement of cognitive functions through cognitive training could facilitate a greater weight loss in patients under hypocaloric dietary treatment.

According to our data, both the cognitive and control groups improved in attention, cognitive flexibility and task planning. Such improvements might be explained by weight loss. In fact, the improvement in cognitive flexibility was essentially similar in both groups. However, only the cognitive group showed a significant improvement in working memory and decision making, which could be due to the effect of cognitive training. In fact, this enhancement coincides with previous data reported in healthy people by Nouchi et al. [26,37]. Furthermore, the cognitive group showed higher post-intervention performance in memory task planning than the controls. Since the cognitive group achieved a higher weight loss, increasing the ability of patients with obesity to organise and plan strategies to solve problems by increasing task planning and working-memory skills might favour weight loss. 

Cognitive training, together with hypocaloric diet, was more effective than just the hypocaloric diet exclusively on the improvement of the BMI, weight loss, body fat percentage and waist circumference. However, the ability of the cognitive intervention to produce changes in lipids or glucose levels in comparison with controls at the biochemical level was modest, probably as a result of baseline values. Also, the lack of differences regarding biochemical parameters may be a consequence of the reduced treatment effect differences, in spite of the statistical significance.

It is important to comment that, in the present study, cognitive training was not targeted for improving food-specific inhibitory control or high-caloric food attentional bias, as previously reported [38,39]. In this study, cognitive training games were not related to food, diet or healthy habits but to logical, mathematical and similar games. Moreover, the withdrawal rate was similar in both groups, which may suggest that this intervention would be suitable for most of the subjects with obesity. To our knowledge, there is only one study determining the effect of training executive functions in an overweight/obese population [40]. Findings from that previous study, conducted with children presenting obesity, are in agreement with our data and support the beneficial effect of cognitive training on weight control [40]. 

The reason why this sort of intervention may be useful in the treatment of obesity relates to the improvement of executive functions. There is some evidence that excess body weight is associated with brain changes and certain cognitive deficits [2,41]. In particular, frontal lobe dysfunction and associated executive deficits have been commonly reported in subjects with obesity [42,43]. Such deficits may mean that these subjects would have problems in adjusting their caloric intake to their energy requirements, due to inadequate planning. 

In obesity, as reported also in psychiatric disorders [44], working memory and decision-making training might improve impulsivity and self-regulation. This behavioural change may be crucial for successful weight control [45]. In this regard, a report by Medic et al. described a significant thinning of the orbitofrontal and the left lateral occipital cortex, associated with an increase in BMI, which suggests a possible shift in reward valuation, goal control and decision processes [46]. Therefore, increasing the ability to interpret and process these cognitive tasks, as is what happened in the present study, could be a starting point in restoring these cognitive processes to a similar functionality than before the BMI increased.

Taking into account our observations and those of previous works, as shown in Figure 4, we hypothesise that there exists a vicious cycle, where several factors interact to promote the development and maintenance of obesity. It is tempting to assume that by improving executive functions, not only we will be able to achieve an effective weight loss, but also, in the long term, will develop a better self-regulation of eating behaviour. Thus, cognitive training, combined with dietary treatment, might be more effective in reducing post-treatment weight re-gain, as it prevents patients from returning to their deleterious habits as soon as they quit the diet. Moreover, the improvement of cognitive deficits may contribute to mitigating the consequences of obesity on health and quality of life.

At this point, several limitations should be commented upon. First, although statistical analysis revealed significant treatment effect differences, there was only a 1% effect difference between groups regarding the primary outcome, therefore, the clinical relevance of the intervention may be limited. Moreover, there was a high drop-out ratio, but similar in both groups. Nevertheless, the high drop-out is somewhat frequent in obesity trials [47]. On the other hand, there is a certain disparity of games for the improvement of executive functions, meaning that other interventions may be also effective [48,49]. In addition, although both groups were advised to increase their daily physical activity, we did not measure adherence to this recommendation. Therefore, the intervention effect may be masked in part by differences in physical activity levels during the intervention. However, at baseline, the International Physical Activity Questionnaire (IPAQ) revealed no significant daily physical activity differences between groups (data not shown). Finally, it is important to remember that the present study was conducted on metabolically healthy individuals; consequently, extrapolation of these results to the general population with obesity should be done with caution.

## 5. Conclusions

To our knowledge, this is the first study to determine the effect of a combined therapy of cognitive training and hypocaloric diet on cognitive and anthropometrical parameters in adults with obesity. Although the data obtained in the present study have shown that the clinical relevance of this intervention was slight, the statistical analysis revealed a significant improvement in all anthropometrical parameters, probably mediated by an improvement in the working memory. Therefore, this kind of cognitive interventions would be helpful in increasing the effectiveness of hypocaloric treatments. Given the shortage of training interventions for executive functions in the context of weight control interventions, this sort of combined intervention could establish the first steps towards a more appropriate intervention for those patients. Further studies will be necessary to improve the cognitive intervention and therefore its effect on anthropometrical parameters.

## Figures and Tables

**Figure 1 nutrients-11-00925-f001:**
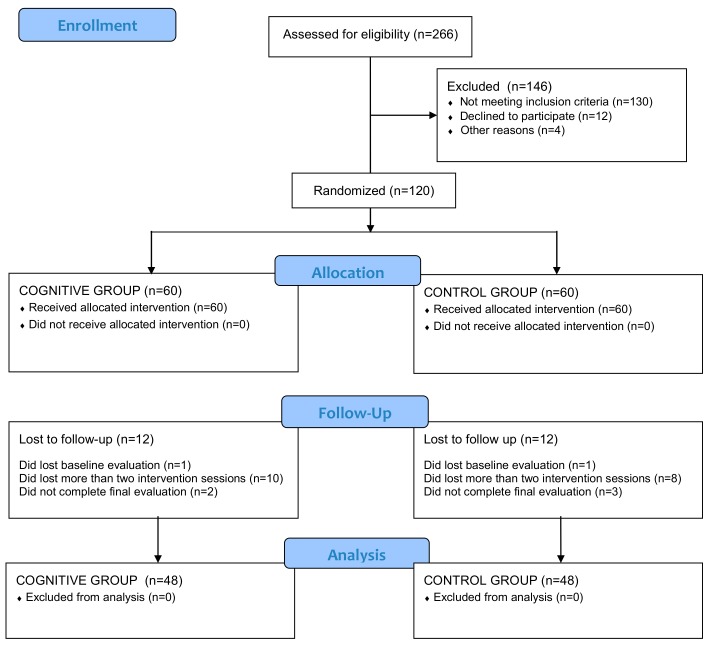
Flow diagram of the trial.

**Figure 2 nutrients-11-00925-f002:**
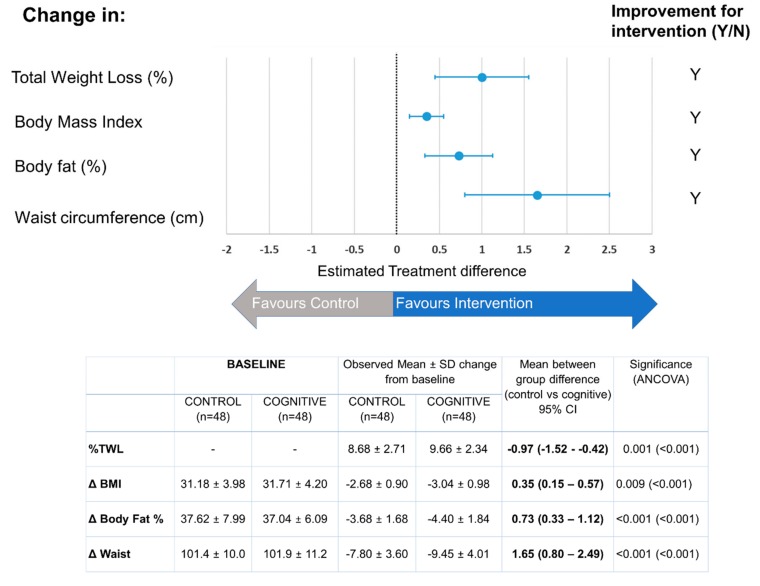
Three-month change estimates for anthropometric variables by treatment. Mean ± SD. Between-group differences were estimated as Mean Cognitive—Mean Control value. A 95% confidence interval of between-group differences is shown in parentheses. Significance was analysed by an ANCOVA test. Significance data, controlling for baseline anthropometric values and sex, is shown in parentheses. Statistical significant differences that are favourable to the cognitive group have been denoted in bold.

**Figure 3 nutrients-11-00925-f003:**
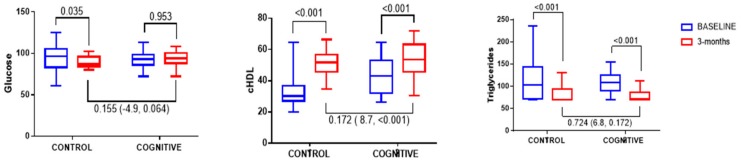
Change in biochemical characteristics after three months of intervention. Significance was analysed by an ANCOVA analysis, controlling for baseline biochemical values and sex. The estimated treatment effect is described in parentheses.

**Figure 4 nutrients-11-00925-f004:**
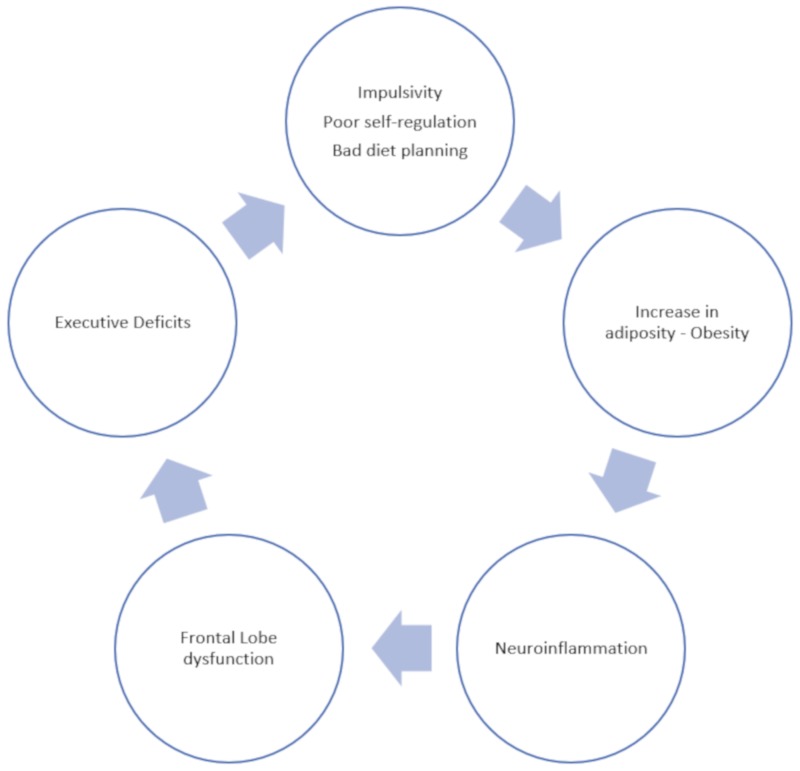
Proposed explanatory model for the relationship between obesity and executive functioning deficit.

**Table 1 nutrients-11-00925-t001:** Baseline participant clinical, anthropometrical and cognitive characteristics.

	CONTROL (*n* = 48)	COGNITIVE (*n* = 48)	Sig. (*t*-test, χ-test)
Age (years)	46 ± 7	44 ± 10	0.376
BMI (kg/m^2^)	31.18 ± 3.98	31.71 ± 4.20	0.643
Waist	101.4 ± 10.0	101.9 ± 11.2	0.856
Body fat (%)	37.62 ± 7.99	37.04 ± 6.09	0.764
SBP (mmHg)	130 ± 13	118 ± 16	0.220
DBP (mmHg)	79 ± 7	71 ± 9	0.120
FPG (mg/dl)	101 ± 23	93 ± 10	0.089
Triglycerides (mg/dl)	117 ± 49	107 ± 27	0.144
cHDL (mg/dl)	34 ± 11	43 ± 11	0.015
Cognitive variables			
Working memory (Z-L-N score)	0.80 ± 1.06	1.04 ± 1.02	0.433
Selective attention (Z-Total d2 test)	0.23 ± 1.52	0.04 ± 1.34	0.656
Z-Focusing Index (d2 test)	−0.66 ± 1.53	−0.70 ± 1.27	0.913
Z-Inhibition (5-digit test)	0.26 ± 1.10	0.17 ± 0.78	0.997
Z-Flexibility (5-digit test)	−0.21 ± 1.36	−0.42 ± 1.01	0.598
Task planning (Z score 30 min)	1.90 ± 0.69	1.96 ± 1.11	0.884
Decision making (Z-IGT score)	−0.82 ± 1.49	−0.31 ± 1.39	0.223

Data represent Mean ± SD. BMI = body mass index, SBP = Systolic blood pressure, DBP = diastolic blood pressure, FPG = fasting plasma glucose, TOT = total d2 test effectiveness, FI = Focusing index, Min = minutes. IGT = Iowa Gambling Task. Cognitive variables were standardized to z-scores.

**Table 2 nutrients-11-00925-t002:** Three-month change estimates for executive function variables by treatment.

	BASELINE		Observed Mean ± SD Change from Baseline		Mean between Group Difference (Control vs. Cognitive) 95% CI	Significance (ANCOVA)
	CONTROL (*n* = 48)	COGNITIVE (*n* = 48)	CONTROL (*n* = 48)	COGNITIVE (*n* = 48)		
Working Memory	0.80 ± 1.06	1.04 ± 1.02	0.23 ± 0.79	0.55 ± 0.69 *	−0.32 (−0.50 – −0.51)	0.001 (0.003)
Selective Attention	0.23 ± 1.52	0.04 ± 1.34	0.69 ± 0.60 *	0.54 ± 0.69 *	0.15 (−0.02 – 0.31)	0.076 (0.054)
Cognitive Flexibility	−0.21 ± 1.36	−0.42 ± 1.01	0.37 ± 0.70 *	0.41 ± 0.79 *	−0.03 (−0.15 – 0.22)	0.693 (0.270)
Task Planning	1.90 ± 0.69	1.96 ± 1.11	0.70 ± 0.66 *	0.96 ± 0.83 *	−0.25 (−0.44 – −0.06)	0.009 (0.003)
Decision making	−0.82 ± 1.49	−0.31 ± 1.39	1.13 ± 2.21	1.23 ± 2.62 *	−0.09 (−0.70 – 0.52)	0.776 (0.603)

Mean ± SD. The data in this table refers to standardised z-scores Between-group differences were estimated as Mean Cognitive—Mean Control value. A 95% confidence interval of between-group differences is shown in parentheses. Significance was analysed by an ANCOVA test. Significance data, controlling for baseline values, age and study level, is shown in parentheses. * represents statistically significant changes after three months regarding baseline values.

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
