# Peer review of "Cognitive Training Therapy Improves the Effect of Hypocaloric Treatment on Subjects with Overweight/Obesity: A Randomised Clinical Trial"

_nutrients, 2019, doi:10.3390/nu11040925_

Reviewer 1 Report

This study aims to assess a potentially effective intervention to assist with weight loss in individuals with obesity. Nonetheless, there are some major aspects of this manuscript that require revision. Please see comments below.

Language

The English language is quite difficult to read throughout the manuscript. I suggest an English review by a native English speaker.

I also suggest using people-first language. For instance, use “people with obesity” instead of “obese patients”.

Abstract

I think that it is important to mention in the abstract that the control group also included cognitive behaviour therapy and nutritional education.

Did the cognitive group include nutrition education? If yes, it is important to mention this.

Page 1, lines 19-20: It seems like the diets were different (i.e., only one type of diet was hypocaloric). Could the authors clarify this in the abstract?

Intervention

Page 4, line 126: Could the authors briefly explain the exercises practiced in the sessions?

Page 4, lines 132-135: Cognitive behaviour therapy is an active intervention, it is not clear for me how this intervention could be only informative. Was cognitive behaviour therapy focussed on assisting with weight loss? Is it possible that cognitive behaviour therapy induced improvements in cognition in the control group?

Figure 2 is a figure and a table? Could these be separated? These are quite difficult to understand, particularly the image. Do the asterisks on the table mean significant change? I suggest presenting the data in a figure or a table (not both) and presenting only the most important information.

Figure 3: again, I suggest presenting the data in a figure or a table (not both) and presenting only the most important information. Also, wouldn’t it be informative to include changes in body weight?

There are many tables/figures. Would it be possible to reduce the quantity of tables/figures?

Results

Page 7, lines 244-247: Visual planning is mentioned here, however this is not included in the image and table below. Is visual planning the same as task planning?

Discussion

Page 9, lines 281-283: These sentences seem disconnected from the study because the aim of the study was not to examine the causes of cognitive deficits in people with obesity.

Page 11, lines 337-339: Was the effect of diet and cognitive training assessed in any long-term follow-up? I do not think this was included in the study. 12 weeks probably cannot be considered a long-term follow-up assessment.

Page 11, lines 342-344: 1% effect difference between groups is probably not significant. From this data, it is not correct to conclude that the cognitive group showed better results than the control group.

Conclusions

With only 0.97% effect difference between groups in weight loss, and between-groups difference only in working memory and task planning is it appropriate to conclude that the cognitive group was more effective than the control group?

Author Response

RESPONSES TO REVIEWER #1

Comments and Suggestions for Authors

This study aims to assess a potentially effective intervention to assist with weight loss in individuals with obesity. Nonetheless, there are some major aspects of this manuscript that require revision. Please see comments below.

Language

#1- The English language is quite difficult to read throughout the manuscript. I suggest an English review by a native English speaker.

We fully agree with the reviewer comment. Therefore, the paper has been reviewed by a professional proofreading service, and several modifications have been performed to try to clarify the reading of the paper.

#2- I also suggest using people-first language. For instance, use “people with obesity” instead of “obese patients”.

We completely agree with the reviewer’s comment. Therefore, we have made this modification throughout the revised paper

#3- Abstract

I think that it is important to mention in the abstract that the control group also included cognitive behaviour therapy and nutritional education.

Effectively, this information is important but was missing in the previous version. We have included this information in the revised version of the paper.

#4- Did the cognitive group include nutrition education? If yes, it is important to mention this.

The comment of the reviewer is fairly interesting, however, there was no educational intervention in the cognitive group. For that reason, there was no mention about this issue in the previous version of the paper. This was because the objective of these sessions was oriented to ensure a similar duration of time and number of sessions than cognitive group, not to increase cognitive performance.

#5- Page 1, lines 19-20: It seems like the diets were different (i.e., only one type of diet was hypocaloric). Could the authors clarify this in the abstract?

We sincerely regret this misunderstanding. The diet was similar in both groups, so, this question has been commented in the revised abstract.

#6- Intervention

Page 4, line 126: Could the authors briefly explain the exercises practiced in the sessions?

Attending to the reviewer’s comment, we have included a brief description of the exercises. Moreover, we have modified supplementary documentation to increase the information about the exercises performed in the training sessions.

 #7- Page 4, lines 132-135: Cognitive behaviour therapy is an active intervention, it is not clear for me how this intervention could be only informative. Was cognitive behaviour therapy focussed on assisting with weight loss? Is it possible that cognitive behaviour therapy induced improvements in cognition in the control group?

This is a good remark by part of the reviewer. Perhaps, the word “informative” is inadequate since, effectively, the intervention was not only informative. Originally, we would like to make reference to the fact that the control intervention was not aimed to improve cognitive functions, as we just report “information” about cognitive-behavioural procedures and nutritional education that might be helpful to the patients. For instance, some of these sessions were oriented to the difference between macro- and micronutrients, the basis of Mediterranean diet and other cognitive-behavioural tips to lose weight, such as increasing physical activity. This intervention was necessary to ensure a similar duration of time and number of sessions than cognitive group, which may have produced a bias in the obtained results.

In our opinion, these sessions were not enough to increase executive or cognitive functions, and to our knowledge, there is no previous work reporting information about this issue. In general, to increase the cognitive performance, specific exercises such as those performed in the present work should be conducted. On the other hand, previous works have observed an increase in several cognitive functions after weight loss[1], so, again in our opinion, the improvement on the cognitive function observed in the control group may be a consequence of the decrease of body weight.

Nevertheless, as the reviewer properly comments, we cannot totally exclude an effect of the cognitive-behavioural intervention on cognitive performance, but we consider that further studies will be necessary to clarify this issue. Therefore, to avoid confusion, we have deleted the sentence “the sessions were only informative” in the revised paper.

References:

1.        Horie, N.C.; Serrao, V.T.; Simon, S.S.; Gascon, M.R.P.; dos Santos, A.X.; Zambone, M.A.; del Bigio de Freitas, M.M.; Cunha-Neto, E.; Marques, E.L.; Halpern, A.; de Melo, M.E.; Mancini, M.C.; Cercato, C. Cognitive Effects of Intentional Weight Loss in Elderly Obese Individuals With Mild Cognitive Impairment. J. Clin. Endocrinol. Metab. 2016, 101, 1104–1112, doi:10.1210/jc.2015-2315.

#8- Figure 2 is a figure and a table? Could these be separated? These are quite difficult to understand, particularly the image. Do the asterisks on the table mean significant change? I suggest presenting the data in a figure or a table (not both) and presenting only the most important information.

We completely agree with the reviewer comment, and therefore, we have decided to delete the figure. In addition, we have included the information about the meaning of the asterisk symbol, which was missed in the previous version of the paper. Effectively, as the reviewer indicates, the asterisks represent statistically significant changes from baseline.

#9- Figure 3: again, I suggest presenting the data in a figure or a table (not both) and presenting only the most important information. Also, wouldn’t it be informative to include changes in body weight?

Although the comment of the reviewer is quite interesting, we consider necessary, at least regarding the primary outcome, to detail the specific data. The forest plot is a usual way to represent the effectiveness of the intervention in clinical trials[2–4], so we consider of great interest to maintain both the figure and the specific data. In addition, we have slightly modified the figure and have changed the sense of the primary outcome to unify criteria and facilitate the interpretation of the figure. Nevertheless, if it will be necessary, the data will be included as supplementary information in a new revised version of the paper.

On the other hand, the data regarding changes in body weight may lead to misinterpretation, since those subjects with higher baseline body weight may lose more weight, but globally, the effect of the intervention may be scarce in these subjects. Therefore, and as it was commented in the paper, several scientific organizations recommend to employ the percentage of total weight loss.

References:

2.           Blackman, A.; Foster, G.D.; Zammit, G.; Rosenberg, R.; Aronne, L.; Wadden, T.; Claudius, B.; Jensen, C.B.; Mignot, E. Effect of liraglutide 3.0 mg in individuals with obesity and moderate or severe obstructive sleep apnea: The scale sleep apnea randomized clinical trial. Int. J. Obes. 2016, 40, 1310–1319, doi:10.1038/ijo.2016.52.

3.           Kolotkin, R.L.; Fujioka, K.; Wolden, M.L.; Brett, J.H.; Bjorner, J.B. Improvements in health-related quality of life with liraglutide 3.0 mg compared with placebo in weight management. Clin. Obes. 2016, 6, 233–42, doi:10.1111/cob.12146.

4.           Blundell, J.; Finlayson, G.; Axelsen, M.; Flint, A.; Gibbons, C.; Kvist, T.; Hjerpsted, J.B. Effects of once-weekly semaglutide on appetite, energy intake, control of eating, food preference and body weight in subjects with obesity. Diabetes, Obes. Metab. 2017, 19, 1242–1251, doi:10.1111/dom.12932.

#10- There are many tables/figures. Would it be possible to reduce the quantity of tables/figures?

We believe that by eliminating figure 2 the number of figures and tables has already been reduced. In addition, the former Figure 4 (biochemical parameters) have been placed on landscape to reduce its size. In any case, as we have commented, it is important to detail the data at least of the primary outcome. Nevertheless, if the reviewer still considers it necessary, we will remove the specific data in a new revised version of the paper.

#11- Results

Page 7, lines 244-247: Visual planning is mentioned here, however this is not included in the image and table below. Is visual planning the same as task planning?

We sincerely apologize for this misunderstanding. Effectively, visual memory task planning made reference to task planning. Then, this question has been corrected in the revised paper.

#12- Discussion

Page 9, lines 281-283: These sentences seem disconnected from the study because the aim of the study was not to examine the causes of cognitive deficits in people with obesity.

We agree with the reviewer, and we have rewritten these sentences to try to clarify this aspect in the revised version of the paper.

#13- Page 11, lines 337-339: Was the effect of diet and cognitive training assessed in any long-term follow-up? I do not think this was included in the study. 12 weeks probably cannot be considered a long-term follow-up assessment.

Effectively, this was a confusion by our part. Therefore, we have decided to delete this paragraph in the revised paper.

#14- Page 11, lines 342-344: 1% effect difference between groups is probably not significant. From this data, it is not correct to conclude that the cognitive group showed better results than the control group.

We fully agree with the reviewer.  As we commented in the original version:

although statistical analysis revealed a treatment effect differences, there was only a 1% effect difference between groups, therefore, the clinical relevance of the intervention may be limited”.

So, in our opinion, although the clinical relevance of this intervention was low or even absent, the statistical analysis revealed an improvement in all anthropometrical parameters, therefore, we consider that the present study may represent a good starting point for further studies including this kind of intervention for obesity treatment.

#15- Conclusions

With only 0.97% effect difference between groups in weight loss, and between-groups difference only in working memory and task planning is it appropriate to conclude that the cognitive group was more effective than the control group?

Again, as in the previous comment, we fully agree with the reviewer. Nevertheless, as commented above, although the difference was very low, we consider that the present work may represent a good starting point to design more appropriate interventions for subjects with obesity, which may ideally include several cognitive exercises to increase several cognitive domains that may improve weight loss. Perhaps, several exercises were more adequate than others, but in our opinion, these aspects should be clarified in further studies.

Nevertheless, we have modified the conclusion in the original paper to reinforce the idea that the effect of the intervention was very modest.

Reviewer 2 Report

Galindo Munoz et al. find that cognitive training combined with a low calorie diet was more effective than low calorie diet alone in reducing body weight, BMI, and adiposity, as well as working memory. This is a well-controlled study that has significant implications on treatment of overweight/obesity.

Although the study is generally well-written and presented, there are a number of issues that need careful attention:

Abstract: line 22, “improved largely” is not an optimal wording. I would leave out largely, as the effects were not that large anyway.

Also, line 26: replace “that” with “than”

Fig. 1: In the boxes: “Discontinued intervention” it should either say “Did lose” or “lost”

Methods: was there any blinding of the experimenters as to the group identity?

Table 1: the percent difference in selective attention and Z-Flexibility are in the order of 80-100%, and it is not clear why such large differences are not significant. Is there an explanation?

Page 7, line 242: Again, delete the word largely, as it is not a “good “word, and the differences are not that large.

Bottom of Fig. 2: Is there an explanation why the baselines values of “Cognitive flexibility” for control (-0.21) and cognitive (-0.42) are so different, almost 100%?

Fig. 3: the unit cm should be added to the delta Waist.

Discussion: line 281; Delete “Today”, and change “obese individuals” to “individuals with obesity”

Line 328: we will be able

Lines 337-338: “diminished in the long run” I did not see any data in the manuscript supporting this statement. Please explain.

Line 339: “must be prolonged, or of a longer duration” should be replaced with: must be dosed higher, or of longer duration.

Line 349: please add “the” between Therefore and intervention effect.

Author Response

Comments and Suggestions for Authors – Reviewer #2.

Galindo Munoz et al. find that cognitive training combined with a low calorie diet was more effective than low calorie diet alone in reducing body weight, BMI, and adiposity, as well as working memory. This is a well-controlled study that has significant implications on treatment of overweight/obesity. Although the study is generally well-written and presented, there are a number of issues that need careful attention:

#1.- Abstract: line 22, “improved largely” is not an optimal wording. I would leave out largely, as the effects were not that large anyway.

#2.-Also, line 26: replace “that” with “than”

#3.-Fig. 1: In the boxes: “Discontinued intervention” it should either say “Did lose” or “lost”

We agree with the reviewer, and in consequence, these changes have been made in the revised version of the paper

#4.-Methods: was there any blinding of the experimenters as to the group identity?

Effectively, the experimenter who designed diets (J.S.G.M.) was blinded to allocation of the subjects. Only the researcher who performed the randomization (J.J.H.M.) and the neurologist who performed the cognitive training (M.G.G.) did know the allocation of the participants.

#5.-Table 1: the percent difference in selective attention and Z-Flexibility are in the order of 80-100%, and it is not clear why such large differences are not significant. Is there any explanation?

#6.-Bottom of Fig. 2: Is there an explanation why the baselines values of “Cognitive flexibility” for control (-0.21) and cognitive (-0.42) are so different, almost 100%?

To the best of our knowledge, the lack of differences was mainly due to the heterogeneity of the performance of the participants. In fact, standard deviation was 3-fold higher than mean value, which limited the presence of statistical differences. Nevertheless, to try to avoid the bias due to baseline cognitive performance, these values were considered as covariates in the ANCOVA analysis. 

#7.-Page 7, line 242: Again, delete the word largely, as it is not a “good “word, and the differences are not that large.

#8.-Fig. 3: the unit cm should be added to the delta Waist.

We thank the comments of the reviewer. We have made such modifications in the revised paper.

#9.-Discussion: line 281; Delete “Today”, and change “obese individuals” to “individuals with obesity”

#10.-Line 328: we will be able

We apologize for these mistakes. We have modified these issues in the revised paper following the reviewer’s indications.

#11.-Lines 337-338: “diminished in the long run” I did not see any data in the manuscript supporting this statement. Please explain.

We apologise for this misinterpretation by our part. Effectively, there was no long-term follow-up data, so we have deleted that paragraph to avoid confusion.

#12.-Line 339: “must be prolonged, or of a longer duration” should be replaced with: must be dosed higher, or of longer duration.

We thank the reviewer’s comment, but, as commented above, we considered more appropriate to delete the complete paragraph to avoid confusion.

#13.-Line 349: please add “the” between Therefore and intervention effect.

Again, we apologize for this grammatical mistake. It has been modified in the revised version of the paper.

Round  2

Reviewer 1 Report

Thank you for addressing my comments/suggestions. The manuscript should be ready for publication now.